# The incidence and mortality of childhood acute lymphoblastic leukemia in Indonesia: A systematic review and meta-analysis

**Dina Garniasih**[1,2,3]*, **Susi Susanah**[4,5], **Yunia Sribudiani**[2,5], **Dany Hilmanto**[4]

**1** Faculty of Medicine, Universitas Padjadjaran, Bandung, Indonesia, **2** Department of Biomedical Sciences, Faculty of Medicine, Universitas Padjadjaran, Bandung, Indonesia, **3** Faculty of Medicine, Universitas Pelita Harapan, Tangerang, Indonesia, **4** Department of Pediatrics, Universitas Padjadjaran, Bandung, Indonesia, **5** Research Center of Medical Genetics, Faculty of Medicine, Universitas Padjadjaran, Bandung, Indonesia

* dina.garniasih@yahoo.com

**Data Availability Statement:** All relevant data are within the paper and its Supporting Information files.

## Abstract

### Background

The incidence of childhood ALL in Indonesia is still largely unknown. The widely mentioned statistics from other countries turn out to be only estimated figures. Other data do not specify the types of leukemia and are not specifically focused on children. Therefore, this study aims to pool incidence and mortality statistics from available studies in Indonesia.

### Methods

We searched five different academic databases, including Pubmed, MEDLINE, Cochrane Library, Science Direct, and Google Scholar. Three Indonesian databases, such as the Indonesian Scientific Journal Database (ISJD), Neliti, and Indonesia One Search, were also utilized. Incidence was expressed as per 100,000 children. We used the Newcastle-Ottawa scale (NOS) to assess the quality of cohort studies. The inclusion criteria are cohort studies published in the languages of English or Indonesian. For this analysis, we define children as 0–18 years old.

### Findings

The incidence rate for childhood ALL was found to be 4.32 per 100,000 children (95% CI 2.65–5.99) with a prediction interval of 1.98 to 9.42 per 100,000 children. The incidence rate is higher in males, with 2.45 per 100,000 children (95% CI 1.98–2.91) and a prediction interval of 1.90 to 3.16 per 100,000 children. As for females, the incidence rate is 2.05 per 100,000 children (95% CI 1.52–2.77) with a prediction interval of 1.52 to 2.77 per 100,000 children. The mortality of childhood ALL ranges from 0.44 to 5.3 deaths per 100,000 children, while the CFR is 3.58% with varying true effect sizes of 2.84% to 4.52%.

**Funding:** The author(s) received no specific funding for this work.

**Competing interests:** The authors have declared that no competing interests exist.

## Interpretation

With 79.5 million children living in Indonesia in 2018, this means that there were roughly 3,434 new cases of childhood ALL. An organized effort between multiple sectors is needed to improve the registries of childhood ALL in Indonesia.

## Introduction

Acute lymphoblastic leukemia (ALL) is a malignancy that is most common in children, especially those aged 1–4 years old [1]. Over the years, through improvements in molecular biology, substantial progress has been made in understanding ALL [1–3]. These advancements can be attributed to sound epidemiological data, such as the incidence and mortality rates of childhood ALL [4]. These epidemiological metrics of ALL frequency form the basis for disease surveillance, healthcare policy formulation and evaluation, and scientific study [5]. We can identify characteristics that explain disparities in incidence and mortality rates by comparing them across studies and nations, which leads to improved disease prevention and treatment information [6, 7].

Non-communicable diseases (NCDs), such as heart diseases, diabetes mellitus, and cancers, are increasingly more widespread than communicable diseases globally, including in Indonesia [8]. From 2012 to 2030, Indonesia will have spent $4.47 trillion on NCDs, with breast cancer alone accounting for 15.7 percent of the total cost. As a result, the expense of all cancers will be significantly higher than the estimated 15.7 percent [9].

Considering this, it will be indispensable to know the incidence rates, prevalence proportions, and mortality rates of childhood ALL to plan healthcare strategies for the future. Yet, the exact incidence rate of childhood ALL in Indonesia is still unknown. A frequently mentioned incidence of childhood ALL is 2.5 to 4.0 new cases per 100,000 children in Indonesia [10, 11]. However, upon tracing the literature, the original source states that the incidence mentioned is only an estimation and that "there was no international or national publication on the incidence of childhood cancer and childhood leukemia in Indonesia" [12]. The next best estimate is achieved from publications that calculated ALL incidence in a single institution, which developed the first institution-based cancer registry in 2000 [13]. Indonesia is the fourth most populous country, with more than 13,000 islands [8]. Therefore, such data on incidence from a single institution might not be enough to represent the incidence of childhood ALL in all of Indonesia.

Indonesia's Ministry of Health did not make childhood cancer a major priority for funding [12]. This is reflected in the lack of precise data on childhood cancer. According to Indonesia's cancer registration system from 2005 to 2007, the incidence of childhood ALL (0–17 years old) is 2.8 per 100,000 children. However, it is not specified whether leukemia refers specifically to ALL [14]. The Global Cancer Observatory (GLOBOCAN) in 2020 found that Indonesia has an age-standardized incidence rate of 5.9–7.3 per 100,000 persons, which puts leukemia as the ninth highest cancer with new cases. However, GLOBOCAN's data does not specify the incidence rate in children either [15].

Gaining information about the epidemiology of childhood ALL in Indonesia is crucial for controlling and managing the disease. Currently, Indonesia is utilizing a national cancer registry called *Sistem Registrasi Kanker di Indonesia* (SriKanDi), but this has not been fully adopted and utilized by all hospitals. The government does not make reporting into this registry mandatory, which may explain why it has not been fully embraced. Some institutions even choose

to develop their own cancer registry, making the national cancer registry unreliable at best [13, 14]. Therefore, the primary aim of this study is to conduct a systematic review and meta-analysis on the incidence of childhood ALL in Indonesia. A meta-analysis has been conducted as individual studies may not have a sufficient sample size [16]. Our secondary analysis calculates the mortality rate and case fatality rate (CFR) of childhood ALL in Indonesia.

## Methods

### Eligibility criteria

The authors adhered to the guidance of the Preferred Reporting Items for Systematic Review (PRISMA) 2020 [17]. The protocol of this review was registered on the International Prospective Register of Systematic Reviews (PROSPERO) database with the registration number CRD42022304184.

The population studied were all children (0–18 years old) diagnosed with acute lymphoblastic leukemia. Molecular and cytogenetics studies were very limited in Indonesia [11]. Therefore, the diagnosis must be confirmed by a minimum of a bone marrow smear with immunophenotyping analysis. The primary outcome of this study is to analyze the incidence rate of childhood ALL in Indonesia with a sub-group analysis of sex, study location, year of the study, and quality of the study. Initially, a sub-group analysis of incidence in each city was planned. However, due to the low number of studies available for each city, we decided to combine them into studies from Javanese and non-Javanese islands. The classification of a sub-group of quality would be based on good versus moderate and poor-quality studies as described below. Lastly, a sub-group analysis of the year when the studies were conducted would be classified into studies before 2010 and those from 2010 onwards. The secondary outcome of this study is to analyze the mortality rate and case fatality rate (CFR) of childhood ALL in Indonesia. There are no intervention or comparator groups.

This review's original inclusion criteria were cohort studies published in English or Indonesian that report incidence rates. However, we realized that we could only find one study that reported an incidence rate [18]. Therefore, we replaced that criterion with cohort studies that included only newly diagnosed ALL patients without time restrictions. We also included grey literature, such as conference abstracts, theses, and dissertations. Those studies must have had enough background information to calculate the incidence rate. The exclusion criteria of this study are reviews, animal studies, and studies that generated negative effect sizes on the lower CI. These negative values indicate that the sample size was too small and could not estimate a stable incidence rate [19]. Therefore, sensitivity analysis would be done if the number of childhood ALL is <50 cases [20]. To ensure literature saturation, citations from review studies were scoured. We also did citation and hand searching to ensure that all available studies were included.

### Search strategy and study selection

The literature search started on 31st December 2021 and ended on the same day. We searched five different academic databases, including Pubmed, MEDLINE, Cochrane Library, Science Direct, and Google Scholar. Three Indonesian databases were also utilized to increase literature saturation: the Indonesian Scientific Journal Database (ISJD), Neliti, and Indonesia One Search. The keywords used were "acute lymphoblastic/lymphoid/lymphocytic leukemia", "incidence", "frequency", "epidemiology", and "Indonesia". The Medical Subject Heading (MeSH) terms for each database can be seen in S1 Table. All records were imported into the Rayyan software, where duplicates were detected automatically and screened manually [21]. This software also allowed authors to collaborate in selecting the relevant studies. Two independent

authors conducted the initial search (SS and DG), importing all the findings into Rayyan software. Another author (YS) cross-checked the initial searches. These three authors independently screened all available studies. Conflicts were resolved by discussion with the experts (DG and DH). In the case of studies with overlapping time points from the same institution, we chose the data that provided us with the most available or most recent.

## Data extraction and quality assessment

Data extraction was carried out independently by one author (DG), then reviewed by a second author (DH) to ensure accuracy. We extracted relevant information, such as study identification (author and year of publication), study characteristics (location and number of years the study was conducted), and participant data (population size, number of male patients with ALL diagnosis, incidence rate, and mortality rate). The data on the child population were extracted from the National Bureau of Statistics [22], and we utilized the mid-point year of the study's implementation. The denominator for the population size was tailored to that particular study period. Therefore, this method ensures that our meta-analysis is not greatly affected by the changing population number. The male and female population data will be used to calculate the incidence of ALL in males and females, respectively.

The Newcastle-Ottawa scale (NOS) was implemented to assess the quality of cohort studies. A score of 7–9 on the NOS implied the study was of good quality, 4–6 indicated a moderate or fair quality, and a score of 0–3 meant that the study was of poor quality [23]. Two of the reviewers (DH and DG) independently assessed the scale, and any discrepancies were addressed with other authors (SS, YS, and DH) until a consensus was attained. If any missing data or further data were needed, corresponding authors were sent an email of inquiry. However, none of the corresponding authors reached back.

## Data synthesis

We computed the incidence by dividing the number of ALL patients by the entire population of that region and expressing it as the number of children per 100,000. If the 95% confidence interval (CI) was not provided, we calculated it using the formula explained by Rutter et al. [24]. We used the inverse variance method to obtain the incidence rate and standard error log, which allowed us to pool the data. In order to adjust the CI of the pooled estimate, a random-effects meta-analysis was employed with the Hartung-Knapp-Sidik-Jonkman method [25]. Heterogeneity would be assessed with prediction intervals [26], and visual interpretations of between-study heterogeneity were explored using a Galbraith plot [27]. Publication bias would be assessed using a funnel plot analysis [28], rank correlation with Begg and Mazumdar's test [29], and Egger's test would be used for a regression intercept [30]. A trim-and-fill analysis would be conducted if there was an asymmetry in the funnel plot [31]. The analysis was conducted using STATA software (Version 17.0, StataCorp, College Station, Texas, USA), and results would be displayed using forest plots.

## Results

We identified 384 manuscripts where 71 of these articles were duplicates. A total of 23 records were eliminated after the title and abstract assessment, with 290 articles sought for a full assessment. A total of 10 articles were included in the analysis. We also found 27 articles from citation and hand searching, which resulted in an additional three manuscripts. Three manuscripts were excluded after sensitivity analysis [32–34]. Thirteen studies were available for systematic review and meta-analysis (S1 Fig). The studies' characteristics and the Newcastle-Ottawa scale scores are presented in S2 Table.

The study period for this meta-analysis is from 1997 to 2019. The incidence rate for child-hood ALL was 4.32 per 100,000 children (95% CI 2.65–5.99) (Fig 1). The prediction interval, which represents an absolute measure of heterogeneity, exhibits an incidence rate spread over a wide area: from 1.98 to 9.42 per 100,000 children. The Galbraith plot indicates no heteroge-neity (S2A Fig), while the funnel plot is asymmetric (S2B Fig). The trim-and-fill analysis sug-gests a significant difference after adjustments (S2C Fig). Begg and Mazumdar's test for rank correlation gives a p-value of 0.2001, indicating no evidence of publication bias. Egger's test for a regression intercept gives a p-value of <0.001, indicating possible evidence of publication bias.

A subgroup analysis of incidence by sex, location, year of study, and quality of study was conducted. As for incidence by sex, 12 studies were available for meta-analysis. One study neg-atively affected the lower confidence interval for the incidence in females and was, therefore, excluded after sensitivity analysis [41]. The incidence rate for males is higher than in females, with an incidence rate of 2.45 per 100,000 children (95% CI 1.98–2.91) and a prediction inter-val of 1.90 to 3.16 per 100,000 children (Fig 2A). As for the females, the incidence rate is 2.05 per 100,000 children (95% CI 1.55–2.55) with a prediction interval of 1.52 to 2.77 per 100,000 children (Fig 2B). The Galbraith plot indicates no heterogeneity for both sexes (S3A and S3B Fig). The funnel plots for the male and female subgroups indicated asymmetries (S4A and 5A Figs). The trim-and-fill analysis suggests no significant difference between the observed and imputed effect sizes for both sexes (S4B and 5B Figs). Both Begg and Mazumdar's test for rank correlation and Egger's test for a regression intercept give a p-value of <0.001 for both sub-groups, indicating possible evidence of publication bias.

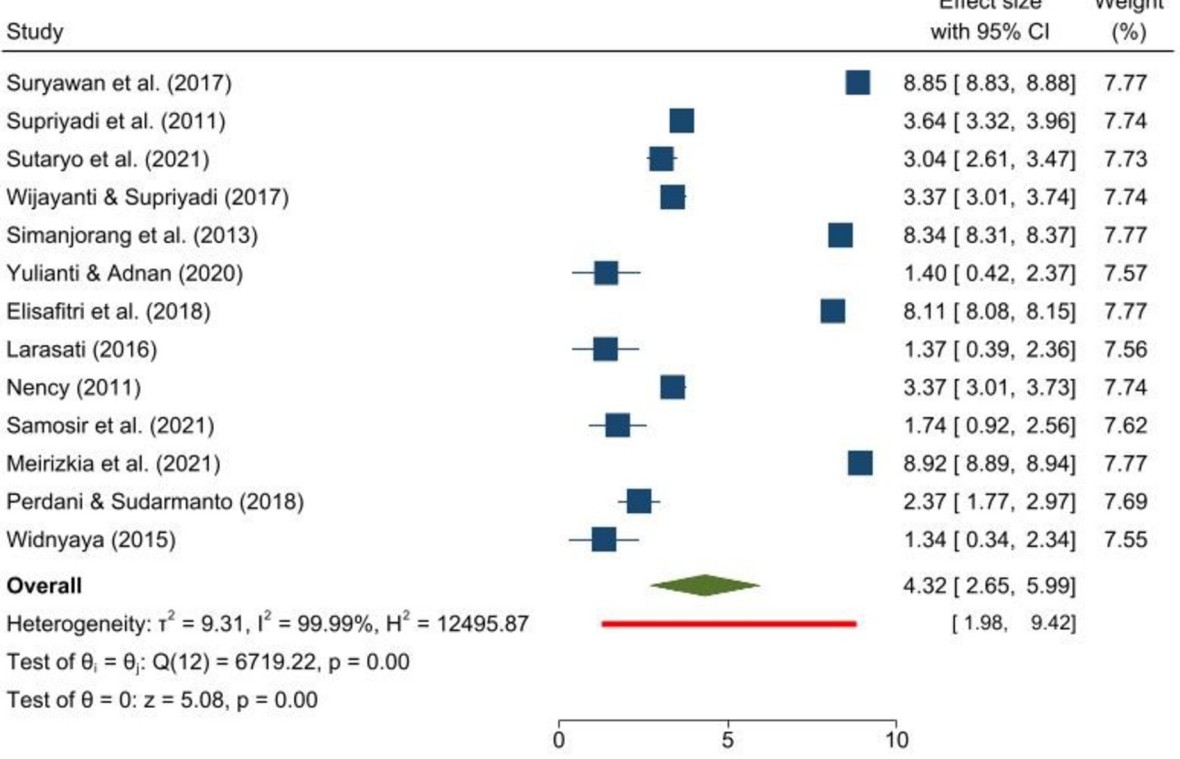

**Fig 1. Forest plot showing the incidence rate and prediction interval of childhood acute lymphoblastic leukemia (per 100,000 children).**

The other subgroup comparison is presented in S3 Table. The incidence of ALL is lower on the Java island than non-Java islands. However, it displays a much narrower prediction interval (an incidence rate of 3.56 on Java Island vs 8.51 on non-Java islands per 100,000 children). Studies conducted before 2010 display a higher incidence rate of 5.12 per 100,000 children (95% CI 1.96–8.29) than studies done after 2010, with an incidence rate of 4.08 per 100,000 children (95% CI 2.07–6.09). However, both studies suffer from a very wide prediction interval. When assessed by the quality, studies of good quality gave an incidence rate of 4.80 (95% CI 2.64–6.99), while moderate quality studies provided an incidence rate of 3.56 (95% CI 0.86–6.25). The prediction interval is narrower in the good quality studies (1.96–11.74) than in the moderate quality studies (0.97–13.14).

We could not perform a meta-analysis on the mortality rate because of the low number of mortalities in almost every study. This negatively affects the lower confidence interval, and almost all studies were excluded via sensitivity analysis. Therefore, the results are presented narratively. Eight studies provided details on the number of deaths in childhood ALL. The mortality rate was highest in a study from Palembang [44], with 5.3 deaths per 100,000 children. Meanwhile, the lowest mortality rate was given in a study in Jakarta [39], with 0.44 deaths per 100,000 children. Three studies showed a mortality rate between 1–2 deaths per 100,000 children per year. These studies originate from Bandung [35], Jakarta [38], and Surabaya [41], with a mortality rate of 1.75, 1.8, and 1.86 deaths per 100,000 children, respectively. The remaining three studies from the Special District of Yogyakarta [36], Makassar [40], and Denpasar [46] have a mortality rate of 2.76, 2.1, and 2.3 deaths per 100,000 children, respectively.

The overall CFR for childhood ALL is 3.58% (95% CI 3.04–4.12) (Fig 3). The prediction interval for CFR is quite narrow: from 2.84 to 4.52%. The Galbraith plot indicates no heterogeneity (S6 Fig). Begg and Mazumdar's test for rank correlation gives a p-value of 0.0094, indicating possible evidence of publication bias. The Egger's test for a regression intercept gives a p-value of <0.001, indicating possible evidence of publication bias.

## Discussion

This study found that the incidence of childhood ALL in Indonesia is 4.32 per 100,000 children. This number is higher than findings from other countries. The incidence of childhood

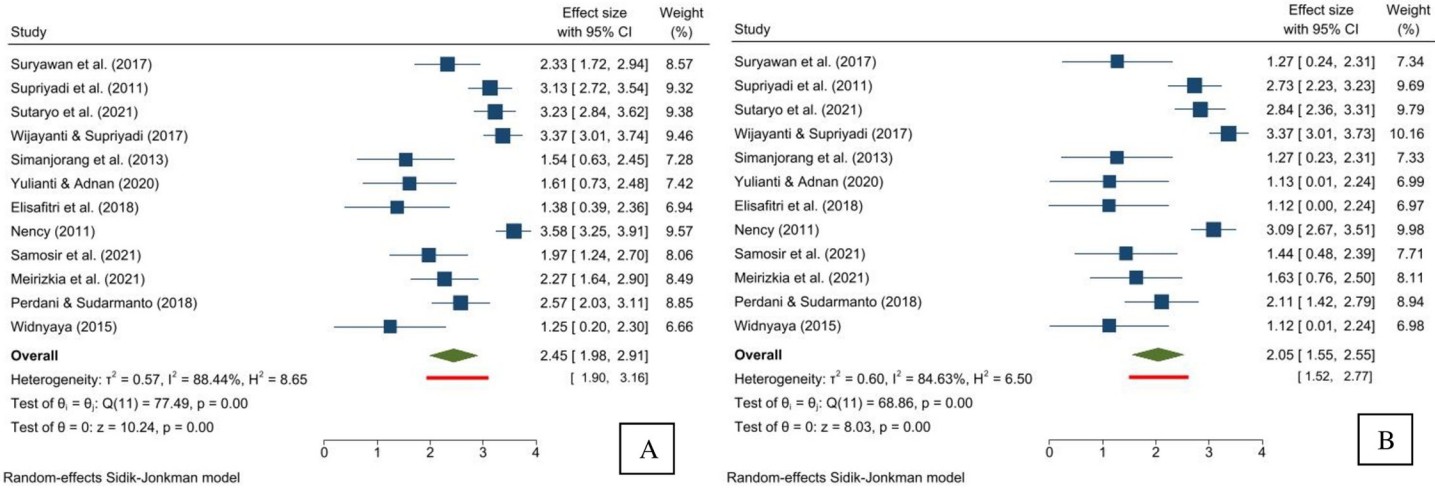

**Fig 2.** Forest plot showing the incidence rate and prediction interval of childhood acute lymphoblastic leukemia (per 100,000 children) for males (A) and females (B).

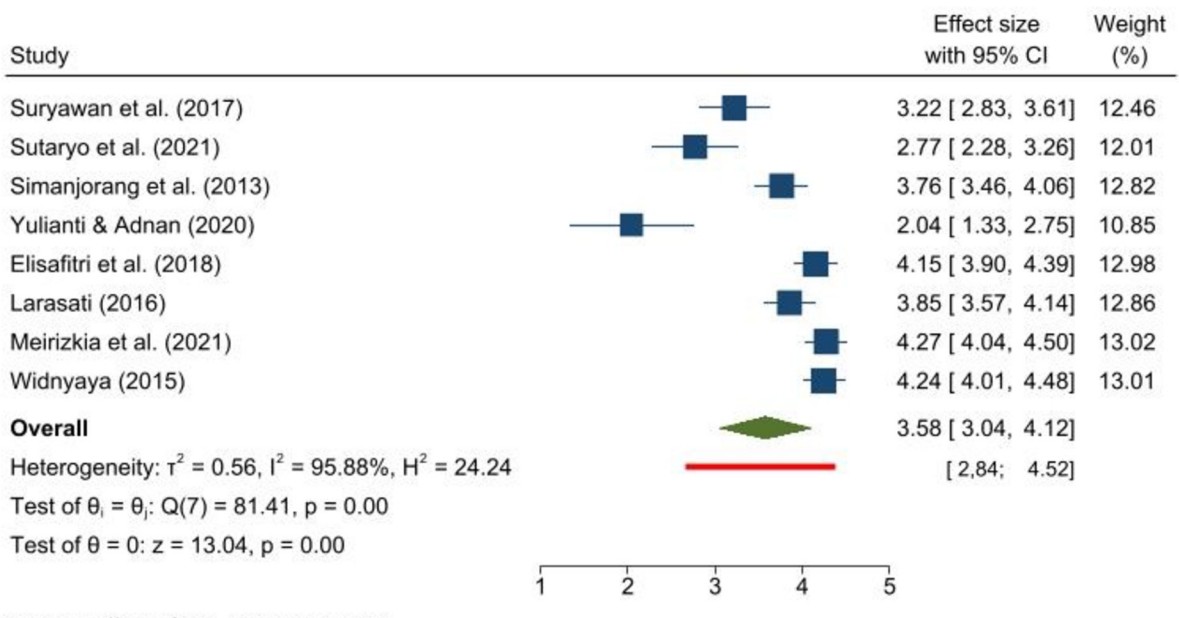

**Fig 3. Forest plot showing the case fatality rate and prediction interval of childhood acute lymphoblastic leukemia.**

(0–14 years) ALL in Nordic countries is 4.0 per 100,000 children per year [47]. In Switzerland, the mean incidence of childhood (<15 years) ALL during four consecutive four-year periods is 3.8 per 100,000 children per year [48]. One study that analyzed Saudi databases found that the age-adjusted incidence of childhood ALL (14 years old or younger) was 2.35 per 100,000 population in 2014 [49]. The Centers for Disease Control and Prevention (CDC) released a finding on the childhood ALL trend from 2001 to 2014. This finding shows that the incidence of childhood ALL is 34 per 1,000,000 people per year in persons <20 years old. However, the incidence can be as high as 75.2 per 1,000,000 people per year in the age group of 1–4 years old [50]. There are three possible explanations for why the incidence rate is higher in our findings. The first is that we include children up to 18 years old, which may contribute to the higher incidence. Secondly, the wide heterogeneity in the true observed effect is depicted by the wide prediction intervals: as the lowest value of the prediction interval is 1.98 per 100,000 children, the true value may be lower or higher than 4.32 per 100,00 children [26]. Indonesia had 79.5 million children [51] in 2018, which means there were roughly 3,434 new cases of childhood ALL. This finding is similar in the United States of America, with >3,000 new cases each year [52]. There is a higher incidence of ALL in males than in females (2.45 vs 2.05 per 100,000 children, respectively). This finding is in concordance with other studies [15, 47, 49, 50]. Lastly, rapid advancements have been made in the ALL diagnostic process, such as the use of genetic and molecular studies. Children who belong to the high-risk groups could be screened and identified earlier, which may contribute to a higher incidence over the years [1–3].

The mortality of childhood ALL ranges from 0.44 deaths per 100,000 children in one study [39] to 5.3 deaths per 100,000 children in another study [44]. The CFR in this study is 3.58%. While children in developed countries have almost 90% cure rates [1, 3], only 20% of children in low- and middle-income countries (LMICs) survive [53]. Refusal or inability to pay for therapy was the most common reason for treatment failure in Indonesia. Treatment-related mortality was the second most common reason for treatment failure [10, 54]. Twinning—an established form of cooperation [55]—between Indonesia and the Netherlands produced an

Indonesian-specific protocol in 2016 named the Indonesian Acute Lymphoblastic Leukemia (ALL) protocol. The development of this protocol has resulted in an estimated survival rate of 60–70% [56]. However, mortality varies greatly by city. Most of the studies included in this review are based on the island of Java. With most of the tertiary center hospitals on that island, 40% of children residing outside Java may not have an equal opportunity for early diagnosis and management [12]. Another plausible explanation for the differing mortality and incidence in Indonesia may be an underreporting of cases.

There are several limitations to this study. The first is the inclusion of studies with small sample sizes. This introduces a systematic sampling error that causes a bias in this finding. We have tried to mitigate this issue with a sensitivity analysis of <50 [57] but, ultimately, some bias will still exist. One of the consequences of this bias is the higher incidence rate before 2010, which seems counterintuitive as, with the further advancement of diagnostic techniques, there should be a better detection rate and, hence, a higher incidence rate [1–3]. Secondly, the funnel plot for incidence shows asymmetry caused by reporting biases, poor methodological quality, true heterogeneity, artefactual, and chance [30]. In terms of reporting biases, we have tried to mitigate this issue by searching Pubmed for ahead-of-print articles and by including all articles, especially grey literature, in English and Indonesian.

Regarding selective outcome and analysis reporting, some studies do not include the full ALL population available due to the inclusion and exclusion criteria [41, 43, 44]. One glaring difference can be found in a study conducted by Samosir et al. [43] in which only 49 children were included out of 495 newly-diagnosed cases of childhood ALL. Therefore, a separate analysis has been done with the original number of ALL cases before the inclusion and exclusion criteria in these three studies. However, the results do not change significantly, and the asymmetry in the funnel plot still exists. Therefore, some publication bias might be at play, despite our best efforts. Egger's test for a regression intercept indicates that the findings suffer from small-study effects, as suggested by our funnel plot that heavily concentrates on the bottom-left corner. Our trim-and-fill analysis gives a lower incidence of 3.0 cases per 100,000 children (95% CI 1.07–4.92). However, the trim-and-fill method's corrected results are not designed to accurately approximate test performance. This method can only be understood as a type of sensitivity analysis for determining the impact of the analysis's bias [58, 59]. The third limitation is that we used a midterm population in our study, which may create inaccuracies in our results. On the same note, some studies specifically excluded children below one year old [39] or included children up to 18 years old. The national statistics only provided population data for 0–4, 5–9, 10–14, and 15–19 years old. Therefore, the denominator for some studies might include children below one year old or young adults of 18–19 years old. However, denominators account for only slight differences in incidence rates and should not significantly distort the results [60].

The fourth limitation is heterogeneity. We did not assess heterogeneity using the conventional $I^2$ proportion because it does not reveal the dispersion of true effect sizes [61]. Cochran's Q test is only used with the DerSimonian-Laird procedure, and therefore we did not utilize this either [62]. Instead, the Galbraith plot was used to visually detect heterogeneity [63] with prediction intervals to tell us how the true effects vary. Although the Galbraith plots do not show any heterogeneity, the prediction intervals show that the true effect sizes vary widely.

The last limitation is related to the situation of childhood ALL in Indonesia. Some children with ALL may die before being given a proper diagnosis or registration in the established cancer registry. The different physical and socioeconomic situations in each region of Indonesia—especially in remote locations outside of Java—could be one of the explanations. Poor families may be unable to access primary health care due to a lack of transportation or financial resources. For instance, a child in a remote area of Jambi (a city in Indonesia) would have to

travel up to eight hours via land transportation to the neighboring city of Palembang just to have access to a pediatric oncologist. If the hospitals there cannot deliver the necessary care, the child would then have to travel from Palembang (Sumatera Island) to Jakarta (Java Island) via land transportation, which could take 2–3 days, or by plane, which would take 45 minutes. The distance traveled is coupled with financial issues for lodging, transportation, and loss of income opportunities, as the parents could not work while accompanying their children [8].

Furthermore, even if the patient could reach a primary health care institution, the medical staff there may not be aware of the symptoms and signs of childhood cancer. There may be a lack of laboratory and other diagnostic facilities, or the parents may be unable to afford necessary tests and treatment [8, 12, 18]. Therefore, the incidence, mortality, and CFR presented in this review may be higher in reality due to underdiagnosis or incomplete registries. Considering all the limitations mentioned above, we can only find a small sample of studies with good quality and sufficient sample sizes, with most studies coming from the Dr. Sardjito Hospital [56].

Despite the limitations, this study is a step forward in encouraging the government, institutions, researchers, and stakeholders to work together nationally and internationally to create a better cancer registry and, hence, more accurate epidemiology of childhood ALL in Indonesia [18, 56, 64]. This is the first systematic review and meta-analysis—to our knowledge—that synthesizes the incidence, mortality, and CFR of childhood ALL in Indonesia. We sincerely hope that, through recognizing the high incidence and mortality of childhood ALL in Indonesia, more funding will be granted for the development of improved epidemiology, diagnostics, and treatments for childhood ALL. The urgency of having a better cancer registry and policies that can provide a reliable source for estimating the incidence and mortality of childhood ALL in Indonesia could not be overstated.

## Conclusion

This study found that the incidence rate for childhood ALL in Indonesia is 4.32 per 100,000 children, with varying true effect sizes of 1.98 to 9.42 per 100,000 children. The mortality of childhood ALL ranges from 0.44 to 5.3 deaths per 100,000 children, while the CFR is 3.58%, with varying true effect sizes of 2.84% to 4.52%. Our findings indicate that the estimation of incidence in childhood ALL needs to be reviewed and improved. A strong and sustained international collaboration has improved and advanced the diagnosis and treatment of ALL in Indonesia. Therefore, twinning needs to be conducted more efficiently and regularly in a larger number of healthcare centers.

## Supporting information

**S1 Table. Medical subject heading (MeSH) terms used in each database.**
(DOCX)

**S2 Table. Characteristics of included studies [18, 35–46].**
(DOCX)

**S3 Table. Incidence of childhood acute lymphoblastic leukemia according to location, year in which the studies were conducted, and quality of studies.**
(DOCX)

**S1 Fig. PRISMA flowchart for selection of included studies.**
(DOCX)

**S2 Fig.** Galbraith plot (A), funnel plot (B), and trim-and-fill analysis funnel plot using R0 (C) of the incidence of childhood acute lymphoblastic leukemia. The observed effect size for the

trim-and-fill analysis is 4.32 (95% CI 2.66–5.99), while the observed and imputed effect size is 3.00 (95% CI 1.07–4.92).
(DOCX)

**S3 Fig.** Galbraith plot of childhood acute lymphoblastic leukemia incidence in males (A) and females (B).
(DOCX)

**S4 Fig.** Funnel plot (A) and trim-and-fill analysis using R0 (B) of the incidence of childhood acute lymphoblastic leukemia in males. The observed effect size for the trim-and-fill analysis is 2.93 (95% CI 2.76–3.08), while the observed and imputed effect size is 3.05 (95% CI 2.90–3.19).
(DOCX)

**S5 Fig.** Funnel plot (A) and trim-and-fill analysis using R0 (B) of the incidence of childhood acute lymphoblastic leukemia in females. The observed effect size for the trim-and-fill analysis is 2.62 (95% CI 2.44–2.80), while the observed and imputed effect size is 2.85 (95% CI 2.68–3.02).
(DOCX)

**S6 Fig. Galbraith plot of the case fatality rate of childhood acute lymphoblastic leukemia.**
(DOCX)

**S1 Checklist. PRISMA 2020 checklist.**
(DOCX)

**S1 Raw data.**
(XLSX)

## Acknowledgments

We would like to thank Gilbert Sterling Octavius for his contribution to preparing this manuscript.

## Author Contributions

**Conceptualization:** Dina Garniasih.

**Data curation:** Dina Garniasih, Yunia Sribudiani.

**Formal analysis:** Dina Garniasih, Dany Hilmanto.

**Investigation:** Dina Garniasih, Dany Hilmanto.

**Methodology:** Dina Garniasih, Susi Susanah, Dany Hilmanto.

**Project administration:** Dina Garniasih.

**Software:** Dina Garniasih, Susi Susanah.

**Supervision:** Dany Hilmanto.

**Validation:** Susi Susanah, Yunia Sribudiani, Dany Hilmanto.

**Visualization:** Yunia Sribudiani.

**Writing – original draft:** Dina Garniasih, Susi Susanah, Yunia Sribudiani, Dany Hilmanto.

**Writing – review & editing:** Dina Garniasih, Susi Susanah, Yunia Sribudiani, Dany Hilmanto.

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
