## [Decision Letter · Decision Letter 0]

1 May 2022

PONE-D-22-09885The incidence and mortality of childhood acute lymphoblastic leukemia in Indonesia: A systematic review and meta-analysisPLOS ONE

Dear Dr. Dina,

Thank you for submitting your manuscript to PLOS ONE. After careful consideration, we feel that it has merit but does not fully meet PLOS ONE’s publication criteria as it currently stands. Therefore, we invite you to submit a revised version of the manuscript that addresses the points raised during the review process.

ACADEMIC EDITOR: Please insert comments here and delete this placeholder text when finished. Be sure to:

The manuscript needs revision please check reviewers comments and address  accordingly

We look forward to receiving your revised manuscript.

Kind regards,

Mohamed A Yassin, MD

Academic Editor

PLOS ONE

Journal Requirements:

2. Please revise the PRISMA checklist such that the corresponding page number and line numbers of the items on the checklist are presented.

4.PLOS requires an ORCID iD for the corresponding author in Editorial Manager on papers submitted after December 6th, 2016. Please ensure that you have an ORCID iD and that it is validated in Editorial Manager. To do this, go to ‘Update my Information’ (in the upper left-hand corner of the main menu), and click on the Fetch/Validate link next to the ORCID field. This will take you to the ORCID site and allow you to create a new iD or authenticate a pre-existing iD in Editorial Manager. Please see the following video for instructions on linking an ORCID iD to your Editorial Manager account: https://www.youtube.com/watch?v=_xcclfuvtxQ

Reviewers' comments:

Reviewer's Responses to Questions

**Comments to the Author**

1. Is the manuscript technically sound, and do the data support the conclusions?

Reviewer #1: Yes

Reviewer #2: Yes

2. Has the statistical analysis been performed appropriately and rigorously? 

Reviewer #1: Yes

Reviewer #2: Yes

3. Have the authors made all data underlying the findings in their manuscript fully available?

Reviewer #1: Yes

Reviewer #2: Yes

4. Is the manuscript presented in an intelligible fashion and written in standard English?

Reviewer #1: Yes

Reviewer #2: No

5. Review Comments to the Author

Reviewer #1: Title:

The incidence and mortality of childhood acute lymphoblastic leukemia in Indonesia: A systematic review and meta-analysis

Outlines:

The authors did a systemic review and meta-analysis from 5 academic databases. They calculated the incidence and mortality of pediatric ALL in Indonesia.

Comments:

This is a well-written article. It provides the updated data of epidemiology of pediatric ALL in Indonesia. I only have one comment for the author.

1. Is it possible to illustrate the trend of incidence and mortality over the study periods? Is the incidence increasing? Is the mortality decreasing?

Reviewer #2: This is an important article and represents an excellent review of data sets to understand the incidence of ALL in Indonesia. There are many obstacles/limitations to this project, which are discussed.

The paper does need an English writer review and rewrite as there are a number of grammatical changes needed.

It is not clear to me what is the time period of all the data reviews. The articles were published from 2007-2021, but not sure how long the time period is and also how this may relate to a changing population size. Has the population size been stable, allowing for an accurate estimate by using the current population size? Some comments to address this would be clarifying.

Can you comment on how global improvements in diagnosis and treatment have affected outcome and also how this led to any increased awareness such as reporting may be increasing over the years (suggesting an increasing incidence).

The connection between cancer funding in Indonesia and how this data will impact access to this funding is not clear.

Would suggest to give a little more info on the local geography as it is implied there are areas of better access, but the common reader may have no understanding of local geography to know how or why this is important.

In the methods, please state how the reviewers defined a quality study.

Can you suggest more why the incidence rate may be higher before 2010, as it seems more intuitive it would be higher later with improved detection?

Give some thoughts as to why the differential mortality rate in different regions. How accurate do you think both the reported incidence and mortality are?

In the discussion one are of limitation is heterogeneity, but it is not clear why this specifically refers to. Heterogeneity of the population by region, age, etc?

The cancer registry is mentioned in the discussion. Comment more about this, when did it start and do all hospitals throughout the regions report to this? Is this required by the government? Will this cancer registry become a reliable source for calculating incidence and mortality?

Finally, how does this work impact a potential improvement in earlier diagnosis, care, and outcomes, as well as documentation for reporting and keeping public records?

6. PLOS authors have the option to publish the peer review history of their article (what does this mean?). If published, this will include your full peer review and any attached files.

Reviewer #1: No

Reviewer #2: No

---

## [Author Response · Author response to Decision Letter 0]

13 May 2022

Manuscript number: PONE-D-22-09885

Article Title: The incidence and mortality of childhood acute lymphoblastic leukemia in Indonesia: A systematic review and meta-analysis

PLOS ONE

Author's Response:

We thank the reviewers for their time and expertise. Please see the specific response to the reviewers enclosed below.

Reviewer(s)' Comments to Author:

Reviewer #1: Title:

The incidence and mortality of childhood acute lymphoblastic leukemia in Indonesia: A systematic review and meta-analysis

Outlines:

The authors did a systemic review and meta-analysis from 5 academic databases. They calculated the incidence and mortality of pediatric ALL in Indonesia.

Comments:

This is a well-written article. It provides the updated data of epidemiology of pediatric ALL in Indonesia. I only have one comment for the author.

1. Is it possible to illustrate the trend of incidence and mortality over the study periods? Is the incidence increasing? Is the mortality decreasing?

Authors' response: Thank you for your valuable input. We did consider charting the trend in incidence and mortality. However, the studies included in this meta-analysis are not suitable for this illustration. This is because while there are multiple studies representing the incidence and mortality in some years, it is not the case for most of the studies. Therefore, we could not extrapolate the incidence from a single period of study in a single institution as the reflection of incidence and mortality in Indonesia and illustrate their trend from that very limited information.

Reviewer #2: This is an important article and represents an excellent review of data sets to understand the incidence of ALL in Indonesia. There are many obstacles/limitations to this project, which are discussed.

The paper does need an English writer review and rewrite as there are a number of grammatical changes needed.

Authors' response: Thank you for your valuable input. We have sent our paper for a professional proofreading check after this comment. We hope it is satisfactory.

It is not clear to me what is the time period of all the data reviews. The articles were published from 2007-2021, but not sure how long the time period is and also how this may relate to a changing population size. Has the population size been stable, allowing for an accurate estimate by using the current population size? Some comments to address this would be clarifying.

Authors' response: Thank you for your valuable input. We have added the study period on line 172 "The study period for this meta-analysis is between 1997-2019". As for the population size, we have delineated that we used the mid-point year of the study's implementation (lines 132-133). However, we have added an extra sentence on lines 138-140, stating, "The denominator for the population size is tailored to that particular study period. Therefore, this method ensures that our meta-analysis is not significantly affected by the changing number of the population" to clarify the sentence further. This means that although the study period is 1997-2019, the population size will remain stable as it is adjusted to that period.

Can you comment on how global improvements in diagnosis and treatment have affected outcome and also how this led to any increased awareness such as reporting may be increasing over the years (suggesting an increasing incidence).

Authors' response: Thank you for your valuable input. We have added a discussion on this topic on lines 248-251 "Lastly, rapid advancements have been made in ALL diagnostics, such as genetic and molecular studies. Children who belong to the high-risk groups could be screened and identified earlier, contributing to a higher incidence over the years(1-3)."

The connection between cancer funding in Indonesia and how this data will impact access to this funding is not clear.

Authors' response: Thank you for your valuable input. We have added a discussion regarding this issue on lines 325-328 "We sincerely hope that after recognizing the high incidence and mortality of ALL in Indonesia as compared to the rest of the world, more funding in childhood cancer is granted in order to obtain a better epidemiological, diagnostic, and treatment for a better future advancement of childhood ALL."

Would suggest to give a little more info on the local geography as it is implied there are areas of better access, but the common reader may have no understanding of local geography to know how or why this is important.

Authors' response: Thank you for your valuable input. We have given an illustration about this issue on line 306-313 "For instance, a child in remote areas of Jambi (a city in Indonesia) has to travel up to a minimum of eight hours via land transportation to the neighbouring city of Palembang, just to get an access to a pediatric oncologists. If the hospitals could not deliver the necessary care, the child has to travel from Palembang (Sumatera island) to Jakarta (Java island) via land transportation which could take up to 2-3 days or plane which could take 45 minutes. The distance travelled is then coupled with financial issues for lodging stay, transportation costs, and opportunity cost lost as the parents could not work while accompanying their children(8)."

In the methods, please state how the reviewers defined a quality study.

Authors' response: Thank you for raising this question. We have defined the quality of a study on line 142-144 "The Newcastle-Ottawa scale (NOS) was implemented to assess the quality of cohort studies. A score of 7-9 on NOS implied the study had a good quality, 4-6 indicated a moderate or fair quality, and a score of 0-3 meant that the study had a poor quality(23)".

Can you suggest more why the incidence rate may be higher before 2010, as it seems more intuitive it would be higher later with improved detection?

Authors' response: Thank you for raising this question. We have attributed this to small sample sizes which could be read furhter on line 270-272 "One of the examples is a higher incidence rate before 2010 which seems counterintuitive as with further advancement of diagnostic techniques, there should be a better detection rate and hence a higher incidence rate(1-3)."

Give some thoughts as to why the differential mortality rate in different regions. How accurate do you think both the reported incidence and mortality are?

Authors' response: Thank you for raising this question. We have actually discussed this on line 255-260 but we have added a sentence on line 265-266 to further clarify this issue "Underreporting of ALL cases may also be a very plausible explanation to differing mortality and incidence in Indonesia". As to the accuracy, we have conceded about this issue on the limitation section on line 267-270 "There are several limitations to this study. The first limitation is the inclusion of studies with small sample sizes. This introduces a systematic sampling error, which causes a bias in this finding. We have tried to mitigate this issue with a sensitivity analysis of <50(57), but ultimately some bias will still exist".

In the discussion one are of limitation is heterogeneity, but it is not clear why this specifically refers to. Heterogeneity of the population by region, age, etc?

Authors' response: Thank you for raising this question. In order to be able to determine where does the heterogeneity comes from, a meta-regression needs to be done which could not be done in our study.

The cancer registry is mentioned in the discussion. Comment more about this, when did it start and do all hospitals throughout the regions report to this? Is this required by the government? Will this cancer registry become a reliable source for calculating incidence and mortality?

Authors' response: Thank you for raising this question. We have added the start of the institution-based cancer registry on line 57 "which develops the first institution-based cancer registry in 2000". Whether all hospitals report to this or required by the government is answered on line 70-74 "Currently, Indonesia is utilizing a national cancer registry called "Sistem Registrasi Kanker di Indonesia (SriKanDi)" but this is not fully adopted and utilized by all hospitals. The government do not make cancer reporting into this registry mandatory, which may explain why it is not fully embraced. Some institutions even choose to develop their own cancer registry, making the national cancer registry unreliable at best(13,14)". As for cancer registry being a reliable source of calculating incidence and mortality is answered on line 329-330 "The urgency for a better cancer registry and policy for a reliable source of estimating incidence and mortality of childhood ALL in Indonesia could not be overstated".

Finally, how does this work impact a potential improvement in earlier diagnosis, care, and outcomes, as well as documentation for reporting and keeping public records?

Authors' response: Thank you for raising this question. We have adddressed this issue on line 321-330 "Despite the limitations, this study is a step forward in encouraging the government, institutions, researchers, and stakeholders to work together nationally and internationally to create a better cancer registry and hence a more accurate epidemiology of childhood ALL in Indonesia(18, 56, 64). This is the first systematic review and meta-analysis, to our knowledge, that synthesizes the incidence, mortality, and CFR of childhood ALL in Indonesia. We sincerely hope that after recognizing the high incidence and mortality of ALL in Indonesia as compared to the rest of the world, more funding in childhood cancer is granted in order to obtain a better epidemiological, diagnostic, and treatment for a better future advancement of childhood ALL. The urgency for a better cancer registry and policy for a reliable source of estimating incidence and mortality of childhood ALL in Indonesia could not be overstated."

We hope that all the reviewers and editor find this response satisfactory. We are more than eager to further revise any comments that are deemed inadequate or necessary for further improvements. Thank you.

---

## [Editor Report · Decision Letter 1]

26 May 2022

The incidence and mortality of childhood acute lymphoblastic leukemia in Indonesia: A systematic review and meta-analysis

PONE-D-22-09885R1

Dear Dr. Dina,

We’re pleased to inform you that your manuscript has been judged scientifically suitable for publication and will be formally accepted for publication once it meets all outstanding technical requirements.

Kind regards,

Mohamed A Yassin, MD

Academic Editor

PLOS ONE

Additional Editor Comments (optional):

The manuscript can be accepted for publication in its current form
---

## [Editor Report · Acceptance letter]

30 May 2022

PONE-D-22-09885R1 

The incidence and mortality of childhood acute lymphoblastic leukemia in Indonesia: A systematic review and meta-analysis 

Dear Dr. Garniasih:

I'm pleased to inform you that your manuscript has been deemed suitable for publication in PLOS ONE. Congratulations! Your manuscript is now with our production department. 

Kind regards, 

on behalf of

Dr. Mohamed A Yassin 

Academic Editor

PLOS ONE